# Progress in Adzuki Bean Seed Coat Colour Studies

**DOI:** 10.3390/plants12183242

**Published:** 2023-09-12

**Authors:** Zhen Wang, Wei Zhao, Yufei Huang, Pu Zhao, Kai Yang, Ping Wan, Liwei Chu

**Affiliations:** 1Schools of Life and Health, Dalian University, No. 10 Xuefu Street, Dalian 116622, China; wangzhen1@s.dlu.edu.cn (Z.W.);; 2Key Laboratory of New Technology in Agricultural Application, College of Plant Science and Technology, Beijing University of Agriculture, Beijing 102206, China

**Keywords:** *Vigna angularis*, genetic analysis, gene mapping, varietal improvement

## Abstract

Seed coat colour is an important quality trait, domestication trait, and morphological marker, and it is closely associated with flavonoid and anthocyanin metabolism pathways. The seed coat colour of the adzuki bean, an important legume crop, influences the processing quality, the commodity itself, and its nutritional quality. In this review, a genetic analysis of different seed coat colours, gene mapping, metabolite content determination, and varietal improvement in adzuki bean are summarized. It provides further insight into gene mapping and cloning of seed coat colour genes and varietal improvements in adzuki beans.

## 1. Introduction

Adzuki bean (*Vigna angularis*) is an important grain legume crop. It was domesticated ~12,000 years ago in China [1] and has been widely cultivated in more than 30 countries, mainly in China, Japan, and Korea [2,3]. Adzuki bean is a traditional Chinese medicinal plant and healthy food. It is widely used in a variety of foods such as paste in pastries, desserts, cakes, porridge, adzuki bean rice, jelly, adzuki milk, ice lollies, and ice cream during traditional celebrations such as the Chinese Spring Festival and Japanese weddings [4,5]. Due to the strong antioxidant activity of its secondary metabolites, adzuki bean is widely used for disease treatment in China [6,7].Adzuki bean is rich in protein, iron, zinc, flavonoids, and vitamin B. Adzuki beans, known as the ‘red pearl’ of legumes, are characterised by their low calorie and fat content, highly digestible protein, abundance of bioactive compounds, and red seed colour [8,9,10]. Seed coat colour is an important domestication trait, commodity-quality trait, and nutrition-quality trait [11]. The biosynthesis of flavonoids causes the change in seed coat colour, which gives adzuki beans their various colours. These flavonoids have medicinal and nutritional value, and antioxidant and antimutagenic activities are associated with their health benefits [12,13]. In adzuki bean, the strong 2,20-azino-bis (3-ethylbenzothiazoline-6-sulphonic acid (ABTSC) free-radical-scavenging capacity is a significant positive correlate with total flavonoid content [14]. Almost all wild adzuki beans (*Vigna angularis* var. *nipponensis*) are black mottle on grey seed coat, and the vast majority of landrace and improved varieties are red. However, landraces come in a wide range of colours: red, black, ivory, black mottle on red, beige, greenish yellow, light brown, brown, green, strong orange yellow, black mottle on grey, and some others (Figure 1).

To date, the information on adzuki bean seed coat colour is still scattered. Focusing on publications of adzuki bean seed coat colour all around the world, this review summarizes the results of genetic analysis, genomic mapping, and metabolite identification of adzuki bean seed coat colour and discusses the molecular regulatory mechanism of adzuki bean seed coat colour. It also suggests future research directions to better understand the regulatory mechanisms between different colours of adzuki bean seeds.

## 2. Genetic Analysis of Seed Coat Colour in Adzuki Beans

Genetic analysis is the first step for researchers attempting to understand the genetic mechanism of different seed coat colours in adzuki beans. According to the investigation results in 1992 and 1993, the seed coat colours of the hybrids between Jingnong2 × S5033 and Jingnong2 × Waiyinye01 are the same as Jingnong2, which is red. The seed coat colour of hybrid S5033 × Jingnong2 is black mottle on red, which is the same as S5033. Seed coat colour in F_1_ hybrids of adzuki beans is controlled by the genes of the female parent [16]. If the seed coat colour is regarded as a qualitative trait and controlled by a single gene, the dominant colour of the parent is generally shown in the F_1_ progeny. The first genetic analysis of adzuki bean seed coat colour in 1917 revealed four genetic loci: red (*R*), green (*G*), and brown colours (*F*) and inhibiting red pigmentation (*H*). However, the original data used in this study are no longer available [17,18]. Crosses of ivory × red and red × light grey showed that the differences between ivory, red, and light grey were controlled by two loci [19]. Reciprocal cross experiments between black mottle on red and red of two F_2_ populations showed that black mottle on red is determined by the dominant allele of a single Mendelian gene [16]. Genetic analysis of the F_2_ and F_3_ populations showed that two loci controlled the difference between the black and red colour of the seed coat. The light brown seed coat colour, which does not occur in the parental lines, appears in the F_2_ individuals. The single Mendelian locus *B* regulates the difference between black and light brown, and another single Mendelian locus *T* regulates the difference between light brown and red. The epistasis of the *B* locus was identified for the first time in this study [20].

Several studies regard seed coat colour as a quantitative trait, and researchers conducted mixed genetic model analyses of multiple loci of adzuki beans. Seed coat colour trait is quantified into three different values. L* stands for lightness; a* stands for redness, from negative (green) to positive (red); and b* stands for yellowness, from positive (yellow) to negative (blue). An early study has indicated that L*, a*, and b* are controlled by one to two, four, and two major genes, respectively [21]. Four F_2_ populations of five parents with different seed coat colours were used for major gene analysis using mixed linear models. The results showed that lightness (L*), redness (a*), and yellowness (b*) were controlled by two major loci. The major effect genes explained 73.31% to 96.88%, 67.62% to 95.03%, and 77.25% to 88.77% of the total phenotypic variance for lightness (L*), redness (a*), and yellowness (b*), respectively [22,23]. In a subsequent study, major genes were reported to explain 78.44%, 38.77%, and 52.18% of the total phenotypic variances of L*, a*, and b*, respectively, which regulate the difference between red and white seed coat colours. The mixed linear model is used in this study. The major gene number is consistent with that of the previous study from Meng [24]. The difference between olive buff and red is analysed as a quantitative trait because the seed coat colour in their F_3_ progeny seemed to vary continuously. The results showed that two loci (*OLB1* and *OLB2*) regulate the difference between olive buff and red, suggesting that *OLB1* and *OLB2* play an important role in suppressing red pigment production. Based on the genetic analysis of F_2_ populations, red seed coat colour difference between ivory, pale olive buff, and red are controlled by quality trait loci *IVY* and *POB*, respectively. Analysis of the F_3_ families derived from self-pollinated F_2_ individuals test the genotype of the F_2_ individuals. Genetic analysis of the F_3_ families showed that the ivory is determined by a recessive allele of the Mendelian *IVY* gene, while the pale olive trait is determined by a recessive allele of a single Mendelian *POB* gene. The two genes act as positive factors for red pigment accumulation [17].

Twelve F_2_ populations and four F_3_ families are used to identified eight genetic loci among nine seed coat colours, and their genetic model was predicted recently (Figure 2) [10]. Gregor Mendel’s laws of heredity and a Chi-square test were used as the main methods of genetic analysis. The twelve F_2_ populations, which were hybridized between red and other seed coat colours, were used to identify the genetic relationship and number of candidate genes between different seed coat colours. Four F_3_ families were used to determine the F_2_ population’s genetic analysis results. The difference between ivory and red was caused by a single *R* locus, which may be the same locus as *IVY* in a previous study [17]. Under the genetic backgrounds of all other recessive loci, the dominant *R* locus causes red seed coat colour, while the recessive *r* locus causes the ivory seed coat colour, conversely. Under the genetic background of a dominant *R* locus, the colours are black mottle on red, brown, golden, light brown, and beige and red, respectively. The difference between black mottle on grey and red is controlled by two loci (*RB* and *BR*), and these differences are all regulated by a single genetic locus including *RB*, *T*, *Y*, *BR,* and *G* loci, respectively. The *B* locus may be the same locus as in a previous study [20] with a dominant epistasis to *Y* and *T* loci [10]. Another study first reported the genetic relationship between light brown and ivory seed coat colours. The results show that the difference between light brown and ivory seed coat colour were controlled by two loci (*R* and *T*). This study first revealed the recessive epistasis of the *R* locus to the *T* locus [25]. The recessive epistasis of the *R* locus to the *G* locus was also reported in a subsequent study [26]. Based on the above conclusion, we can infer that the *R* locus is the recessive epistasis to most other loci. Moreover, the *R* locus is considered to cause the stop of anthocyanidin synthesis in adzuki bean seed coats. However, further genetic analysis is needed to verify this inference.

The above studies greatly enrich the knowledge of genetic relationships between different seed coat colours of adzuki beans. There are different opinions on whether adzuki bean seed coat colour should be considered a quantitative or qualitative trait. There are differences in the conclusions between studies that analyse seed coat colour as a quantitative trait. But results of Gregor Mendel’s law of heredity and the Chi-squared test in different studies can confirm one another. It means that the difference between two seed coat colours in adzuki beans should be considered a qualitative trait rather than a quantitative trait for genetic analysis. The difference between two seed coat colours can be controlled by one or two genes. Interestingly, there are multiple epistasis situations among these genes. Further mapping and functional verification of these genes that regulate these epistasis situations are needed. These conclusions will further improve the gene mapping of adzuki bean seed coat colour. However, it is still necessary to perform complete diallel hybridization on all seed coat colours of adzuki beans to construct a complete genetic relationship network of adzuki beans’ seed coat colours.

## 3. Gene Mapping of Seed Coat Colour in Adzuki Beans

Based on the results of the genetic analysis, gene mapping is used to identify candidate genes for different colours of adzuki bean seed coat. Adzuki bean was first predicted to have an 11 LG genome of 539 Mb [27]. However, the mapping of adzuki bean seed coat colour was barely carried out before the first genetic linkage map was drawn up. This linkage map with a total length of 832.1 cM is constructed from a (*Vigna nepalensis Tateishi & Maxted.* × *Vigna angularis*) × *Vigna angularis* backcross population consisting of 187 individuals. Totals of 205 SSR markers, 187 AFLP markers, and 94 RFLP markers were mapped onto 11 linkage groups corresponding to the haploid chromosome number of adzuki bean [28]. This genetic linkage map opens up the gene mapping of adzuki beans. The first mapping study on seed coat colour used a BC_1_ F_1_ population composed of 187 individuals and an F_2_ population composed of 141 individuals. Presence or absence of black mottle on seed coats and seed coat colour differences between tan and red are both treated as qualitative traits. The genetic analysis of the BC_1_F_1_ population and the F_2_ population shows that the two traits are each controlled by one locus. The mapping identifies SDC in linkage group 1 (LG 1) as controlling the difference between red and tans, and SDCBM in linkage group 4 (LG 4) as controlling the presence or absence of black mottle, respectively [29]. Mapping in the F_2_ population of 188 individuals showed that the *sdcbm3.4a.1* gene, which may control the presence or absence of black mottle on the seed coat, is closely linked with the SSR marker CEDG185 in LG 4b. The gene *sdc 3.1.1*, which may control the difference between the red and ivory seed coat colours, is located near the SSR marker CEDG053 on LG 1 [30]. In this study, the seed coat colour traits are also treated as qualitative traits.

The seed coat colour difference between olive buff and red is first regarded as three quantitative traits including L*, a*, and b* for QTL mapping. The results of the QTL analysis showed that L*, a*, and b* values were determined in the same single 20.0 cM region located between the CEDG141 and CEDG001 SSR markers at LG 1. This region is designated as the olive buff1 (*OLB1*) gene. This major gene explained 54, 43, and 56% of the total phenotypic variances in lightness, redness, and yellowness, respectively. The Acc2265 allele at *OLB1* increased lightness and yellowness, but it reduced the redness of the seed coat. And the value of a * also depends on a second QTL located in the SSR marker CEDG 214, which is 80 cm away from *OLB1*. This region is called the olive red buff2 (*OLB2*) gene, which can only explain 6% of the total phenotypic variation in redness. Based on the genetic analysis results, the *IVY* and *POB* are mapping as qualitative traits. Of the 196 SSR markers, 63 (32.1%) were polymorphic between the red parents and ivory parents. *IVY* is mapped in an interval between SSR markers CEDG016 and CEDG059 on LG 8. The CEDG092 and CEDG112 are molecular markers with the closest genetic distance to *IVY*. Among the 196 SSR markers, 53 (27.0%) were polymorphic between pale olive buff and red parents. The POB is associated with two SSR markers on LG 10, CEDG081 and CEDG116, at distances of 36.1 cM and 34.9 cM, respectively. The most probable linkage sequence is CEDG 116-CEDG 081-POB. In brief, olive buff1 (*OLB 1*), olive buff2 (*OLB 2*), ivory (*IVY*), and pale olive buff (*POB*) were located in linkage groups (LGs) 1, LG 1, LG 8, and LG 10, respectively [17]. Another promising finding in this study is that it seems feasible to regard seed coat colour as a quantitative trait for gene mapping. In the above studies, the gene mapping of adzuki beans remains at the level of linkage groups and genetic maps.

With the publication of a high quality adzuki bean genome draft [31,32,33], the gene mapping level of adzuki beans has been improved from the genetic linkage group level to the chromosome level. The genome draft reported in this study strengthens the correspondence between the genetic linkage map and the physical map of adzuki bean. Based on the new genome, we constructed a high-density genetic linkage map and mapped two genes that regulate the differences between red and black seed coat colours to the tops of chromosomes 1 and 3, respectively. *VaScR,* which may correspond to the *T* locus, is mapped between s92-2653060 and the top of chromosome 1. But no further study has been conducted to map it. *VaSDC1* (*VaScb*) corresponds to the *B* locus, and it is first located in the 131,943–133,424 bp interval between the scaffold326-733037 and the top of chromosome 3 [34]. *VaSDC*1 is regard as a morphological marker *SDC1*, which is further fine mapped between the SSR markers Sca326-12 and BAgs007 at distances of 4.3 cM and 3.1 cM, respectively. The physical distance between Sca326-12 and BAgs007 is 1,271,141 bp. This gene is identified as a R2R3-MYB transcription factor, which displays the highest homology with *AtMYB75* influencing the colour of *Arabidopsis thaliana*. During the three different colouring stages of seed development, *VaSDC1* was specifically expressed in the black seed coat, which may activate structural genes in flavonoid metabolism and cause the differences between black AG118 and red Jingnong6 [35]. Together, the above finding first reveals the molecular mechanism in which a single genetic locus regulates the difference between two seed coat colours in adzuki bean.

Among the gene mapping study of adzuki bean seed coat colour, most researchers regard the seed coat colour trait as a qualitative trait and map the morphological markers to the genetic linkage map. This gene mapping method is mainly due to the results of genetic analysis. However, there is also a study that provides another idea for seed coat colour gene mapping in which we can decompose seed coat colour into three quantitative traits L*, a*, and b* [17]. The molecular markers and candidate genes enrich our understanding of molecular mechanisms in adzuki bean seed coat colours and will lead to the improvement of seed coat colour breeding in adzuki beans (Table 1). However, further genetic localization is still needed to enrich the knowledge of adzuki bean seed coat colour’s molecular regulation mechanism.

## 4. Flavonoids and Pigment Composition of Seed Coat Colour in Adzuki Beans

The identification of key pigments is also an important part of the study on adzuki bean seed coat colour. Anthocyadin is usually considered the main pigment in the seed coat of adzuki beans. The first report of a pigment identified in the seed coat of adzuki bean named 3-monoglucoside delphinidin in the black-red seed coat colour of adzuki beans [36]. The most landrace and improved varieties of adzuki bean are red seed coat varieties. Yoshida isolated an anthocyanin 3-O-(β-D-glucopyranosyl)-5-O-(β-D-glucopyranosyl) cyanidin and regarded this cyanidin as the main pigment of red adzuki beans for the first time [37]. But his further study proved that this anthocyanin (cyanidin 3,5-di-O-glucoside) may not be the key pigment of red adzuki bean seed coats [37,38]. Two ethyl acetate-soluble purple pigments were extracted from the red seed coat of adzuki beans and identified by HPLC, LC-MS, and NMR. Pigment 1 is the product of the condensation of cyanidin and (+)-catechin, in which the 5-hydroxy and C-4 positions of the cyanidin molecule are modified by the addition of the 5-hydroxy and C-6 positions of the (+)-catechin molecule, respectively. Pigment 2 is an isomer of pigment 1. These two pigments may be able to interact with starch hydrophobically and cause the difficulty of adzuki bean pigment degradation in cooking [39,40].

Three anthocyanins such as cyanidin-3-glucoside, pelargonidin-3-glucoside, and pelargonidin-3-sambubioside are identified at a content of 6.7 mg per 100 g of fresh weight [41,42]. According to Sreerama et al. [43], the anthocyanin content of red bean seed coat ranges from 3.14 to 7.94 mg/g dry matter (mg cyanide-3-glucoside equivalent per g of defatted meal). But the anthocyanin content in one study was much higher than that in previous study. The content of total anthocyanins in samples AEP-1 and AEP-2 were 97.3 and 27.8 mg/g dry weight, respectively. In addition, peonidin-3-rutinoside and malvidin-3-O-glucoside have been found in adzuki bean extract [44]. The methanol (MeOH) extract of adzuki beans gave two proanthocyanidins as possible red pigments. Among them, structural elucidation of compound 14 revealed a glyoxylic acid-bridged catechin skeleton to be a key partial structure for forming the branched skeleton of adzuki bean pigments. The presence of compound 14 was important for elucidation of the mechanism of adzuki bean seed coat colour development in proanthocyanidins [45]. Catechinopyranocyanidins A and B were considered to be important purple pigments of red adzuki bean seed coats (Figure 3) [38]. According to this study, the seed coat pigments of red adzuki bean seeds are not anthocyanins, but hydrophobic catechinopyranocyanidins A and B. This result reasonably explains the technological procedure of ‘an-paste’ dyeing by boiling in water and repeated rinsing with water. Another study used a pH differential method [46] after slight modification to measure the total anthocyanin content. The difference in absorbance between A510 and A660 at pH 1.0 and pH 4.5 was used to estimate the concentration of anthocyanin in the samples; the result shows that anthocyanin is barely detectable in the red seed coat of adzuki bean, but this result, which is in contrast to the results of previous study, needs further verification [47].

Black is another important colour of adzuki beans, which is rich in anthocyanins with antioxidant activity. Delphinidin 3-O-glucoside was isolated from the black adzuki bean [36]. To further isolate and identify the anthocyanins in the black adzuki bean (*V. angularis,* cv. Geomguseul) reverse phase Medium Pressure Liquid Chromatography (MPLC) and Ultra Performance Liquid Chromatography (UPLC)/Orbitrap-MS analysis is used, respectively. The isolated anthocyanins are characterised as ten anthocyanin derivatives in black adzuki bean seed coats as delphinidin-diglucosede, delphinidin-3,5-diglucoside, delphinidin-3-rutinoside, delphinidin-3-coumaroylglucoside, cyanidin-3-glucoside, petunidin-3-galactoside, petunidin-3-glucoside, and petunidin-3-(6″-coumaroyl) glucoside. The key anthocyanins delphinidin-3-O-galactoside and delphinidin-3-O-glucoside of delphinidin type were isolated using reversed phase C-18 MPLC. The black adzuki bean can be considered a good source of natural antioxidants and can be used as a functional and healthy food. This study is the first of this kind of research to ascribe the antioxidant ability of the black adzuki beans to ABTS and DPPH free radicals [48]. The difference in adzuki bean seed coat colour between red (Jingnong6) and the black (AG118) is caused by the accumulation of four kinds of cyanidin derivatives and five kinds of delphinidin derivatives including cyanidin O-syringic acid, peonidin O-hexoside, cyanidin 3-0-glucoside, cyanidin, petunidin 3-0-glucoside, malvidin 3-O-glucoside, malvidin 3-O-galactoside, malvidin 3,5-diglucoside, and delphinidin 3-0-glucoside [35]. Based on the previous study, delphinidin derivatives seemed more likely to be the key pigment in black seed coat colour adzuki beans, although cyanidin derivative and delphinidin derivative contents are different between red and black seed coat colours of adzuki beans.

Different colours of adzuki bean seed coats have different pigment compositions, and the combination of procyanidins and anthocyanins affect seed coat colour, and there are no carotenoid or pelargonidin derivatives in the seed coats of any accessions. Therefore, it is speculated that the simpler the colour phenotype of adzuki bean, the fewer anthocyanins are accumulated in its seed coat. The green seed coat colour of adzuki beans is caused by the accumulation of chlorophyll, and anthocyanins may not be accumulated in the seed coat of ivory adzuki beans [15]. Procyanidin is also an important pigment in adzuki bean seed coat colour [43,49]. The previous articles showed that adzuki beans contain flavonoids, such as (+) epicatechin, (+) catechin, quercetin, and vitexin, or their derivatives, with strong antioxidant activity [50,51]. The main objective of one study was to identify and evaluate the antitumour activity of proanthocyanidin-enriched fractions of the adzuki beans. To the best of our knowledge, it is the first study on characterisation of proanthocyanidin-enriched fractions from adzuki beans, which have structures ranging from epicatechin to trimer to nonamer and can contain the pyrogallol moiety. The HPLC results for the ethyl acetate-eluted fraction from the Amberlite XAD-1180N resin identify catechin glucopyranoside, (epi) catechin dimer, procyanidin B4, arecatannin A1, arecatannin A2, (epi) catechin nonamer, piceid, and dehydroquercetinrhamnoside [52].

## 5. Conclusions and Outlooks of Seed Coat Colour in Adzuki Beans

Varied flavonoids and anthocyanins also accumulate in the seed coats of adzuki beans as the main colouring pigments, which result in the colourful seed coats of adzuki beans [15]. Flavonoids and anthocyanins extracted from adzuki beans are related to the antioxidant, immune-regulatory, and radical-scavenging biological activities of adzuki beans [14,53]. Therefore, seed coat colour is not only an important domestication trait, commodity-quality trait, and nutrition-quality trait but is also related to its biological activities. So far, a large number of studies have been conducted on seed coat colours of adzuki beans. Results of genetic analysis, gene mapping, and metabolite determination have improved the seed colour breeding for adzuki beans. However, the molecular regulation mechanism among the complex seed coat colours of adzuki beans is still poorly understood. Previous studies mainly focused on black and red adzuki beans, but little evaluation was made of the physical and chemical properties of adzuki beans with other colours. In addition, in many previous studies, the seed coat colour was not determined, which makes it difficult to evaluate the relationship between the seed coat colour and the characteristics of accessions with different seed coat colours [54]. It is expected that the molecular regulation mechanism between different seed coat colours of adzuki beans needs to be further studied. In addition, the physical and chemical properties of different seed coat colours of adzuki beans will be studied more in the future.

## Figures and Tables

**Figure 1 plants-12-03242-f001:**
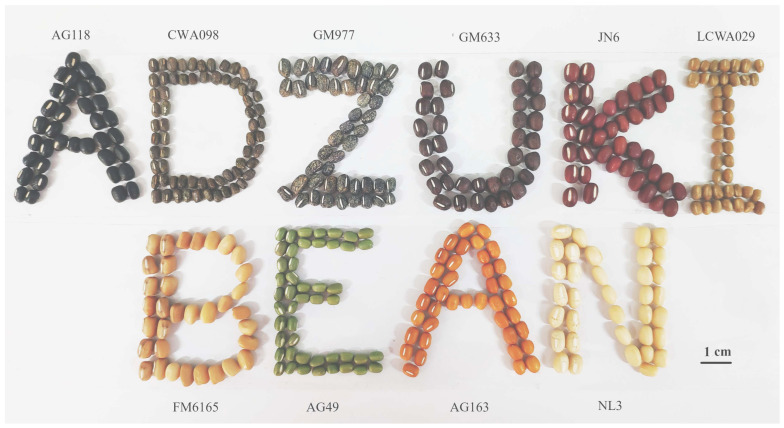
Ten kinds of seed coat colours in adzuki bean [15]. Jingnong6 (JN6), a red improved variety; GM633, a red landrace with black mottle; GM977, a black mottle on grey landrace; AG163, a golden landrace; AG49, a green landrace; AG118, a black landrace; Norin3 (NL3), an ivory improved variety; CWA098, a brown wild accession with black mottle; LCWA029, a brown landrace; and FM6165, a light brown mutant of JN6, are some of the differently coloured varieties.

**Figure 2 plants-12-03242-f002:**
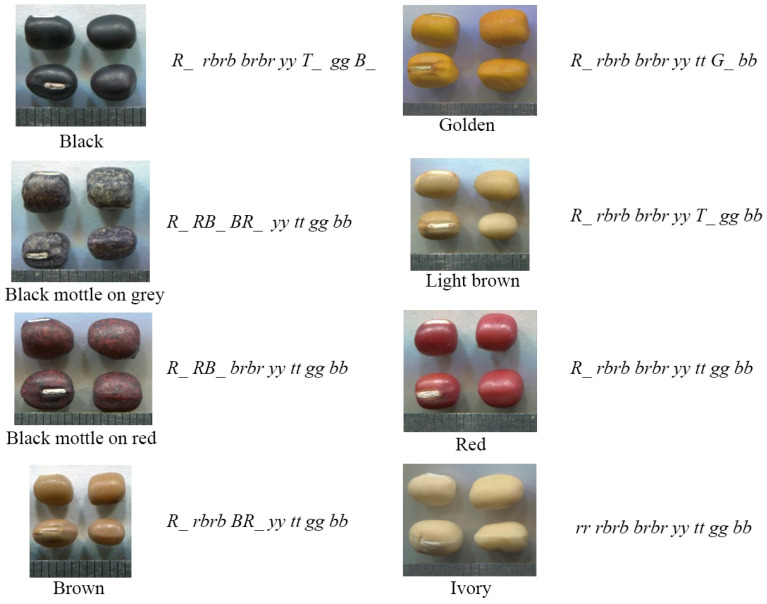
Genetic relationships of seed coat colour loci in adzuki beans [10].

**Figure 3 plants-12-03242-f003:**
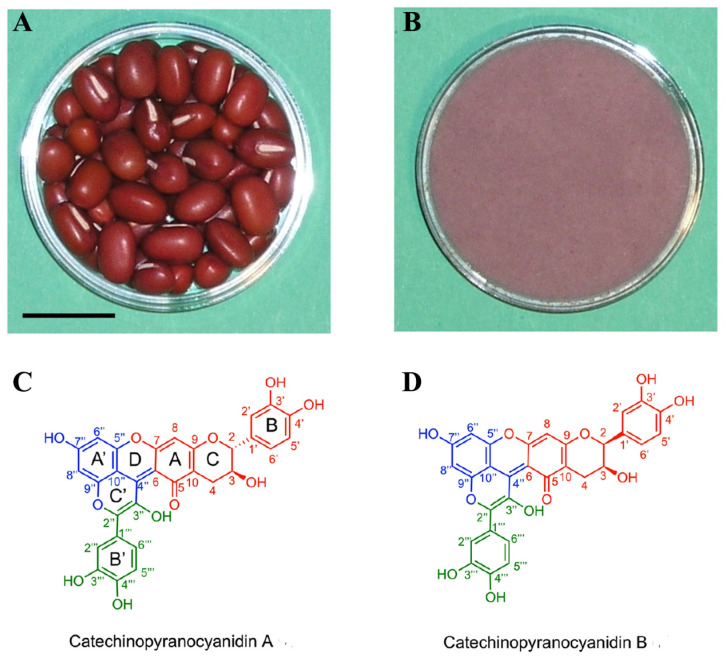
Structures of catechinopyranocyanidins from the red seed coats of adzuki beans [38]. (**A**): Red seed coat adzuki bean (scale bar: 10 mm); (**B**): the paste from seed coat adzuki bean. (**A**) and (**B**) come from Figure 1 of ref [38] and have been modified. (**C**): structure of catechinopyranocyanidin A; (**D**): structure of catechinopyranocyanidin B. (**C**) and (**D**) come from Figure 3 of ref [38] and have been modified. (The 1,2,3,5-tetrasubstituted benzene ring is indicated in blue; 1,3,4-trisubstituted benzene ring is indicated in green; B–C rings of the catechin residue are indicated in red).

**Table 1 plants-12-03242-t001:** Genetic loci in adzuki bean seed coat colour studies.

*Loci*	Seed Coat Colour	Gene	Chromosome/ Linkage Group	Description	References
*OLB1*	Red-olive buff		LG 1	Inhibition of red pigment production.	[17]
*OLB2*	Red-olive buff		LG 1	Inhibition of red pigment production.	[17]
*POB*	Red-pale olive buff		LG 10	A positive factor in red pigmentation.	[17]
*B*	Black-light brown	*VaSDC1*	Chr 3	High expression of this gene activates the flavonoid metabolism pathway’s structural gene expression, promotes the accumulation of anthocyanins, and leads to the differences between black and red seed coat colour of adzuki beans.	[10,35]
*T (SDC)*	Light brown-red	*sdc3.1.1*	LG 1	Regulates the difference between light brown and red seed coat colour.	[10,30]
*RB(SDCBM)*	Back mottle-no mottle	*sdcbm3.4a.1*	LG 4	Regulates the difference between seed coat colour with mottle and without mottle.	[10,30]
*R(IVY)*	Red-ivory		LG 8	Regulates the difference between red and ivory seed coat colour.	[10,17]
*BR*	Brown-red			Regulates the difference between brown and red seed coat colour.	[10]
*G*	Green-red			Regulates the difference between green and red seed coat colour.	[10]
*Y*	Beige-red			Regulates the difference between beige and red seed coat colour.	[10]

## Data Availability

No new data were created or analyzed in this study. Data sharing is not applicable to this article.

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
