# Peer review of "Progress in Adzuki Bean Seed Coat Colour Studies"

_plants, 2023, doi:10.3390/plants12183242_

Round 1

Reviewer 1 Report

The reviewed paper is a review on genetics and biochemistry of seed coat color in adzuki bean. The significant part of this text covers previous papers of the same team of authors. In my opinion, this paper cannot be accepted for publication in its present form. Here are major concerns about this work.

1. First of all, this text requires thorough revision considering its language and style. Authors are strongly recommended to have their paper checked either by a native speaker or by a specialized language service. In many places, I did not manage to understand what authors mean, so before the second round of review the language of this paper should be improved.
2. The chapter devoted to genetic control of seed coat color is almost impossible to understand. It is strongly recommended to carefully revise all terms describing this control. The best solution is to make a table in which all possible genotypes with corresponding phenotypes would be listed. In a form of a text, this information is very difficult to appreciate. Figure 2 is also not a scheme of genetic control. It is impossible to rule out what arrows indicate, why they branch dichotomously, why they fuse etc.
3. The discussion of molecular mechanisms of seed coat formation is very difficult to read; it should be given as a scheme or a table instead. What is more, any discussion like this is impossible without comparison with related species, especially Phaseolus vulgaris, which also exhibits a broad range of seed coat color variation but studied much more thoroughly, with many genes already identified. Key papers on seed coat color in P. vulgaris need to be covered in this review.

Many more corrections, suggestions and questions are available in the paper file (see attached). I hope these might be helpful. Please note that my comments on language and style are not exhaustive, there should be a serious work on text revision.
I recommend to reject this paper with an encouragement to resubmit it after very deep elaboration. I also wish authors good luck.

This paper requires very serious correction considering its language and style. 

Author Response

The reviewed paper is a review on genetics and biochemistry of seed coat color in adzuki bean. The significant part of this text covers previous papers of the same team of authors. In my opinion, this paper cannot be accepted for publication in its present form. Here are major concerns about this work.

  1. First of all, this text requires thorough revision considering its language and style. Authors are strongly recommended to have their paper checked either by a native speaker or by a specialized language service. In many places, I did not manage to understand what authors mean, so before the second round of review the language of this paper should be improved.

Response: Thank you very much. We have improved our manuscript.
2. The chapter devoted to genetic control of seed coat color is almost impossible to understand. It is strongly recommended to carefully revise all terms describing this control. The best solution is to make a table in which all possible genotypes with corresponding phenotypes would be listed. In a form of a text, this information is very difficult to appreciate. Figure 2 is also not a scheme of genetic control. It is impossible to rule out what arrows indicate, why they branch dichotomously, why they fuse etc.

Response: Thank you very much. We have improved the chapter devoted to genetic control of seed coat colour and there is a table in our paper. Figure 2 clearly explained the genetic relationship of eight adzuki bean seed coat colour in our paper before and we have added some note in this paper.
3. The discussion of molecular mechanisms of seed coat formation is very difficult to read; it should be given as a scheme or a table instead. What is more, any discussion like this is impossible without comparison with related species, especially Phaseolus vulgaris, which also exhibits a broad range of seed coat color variation but studied much more thoroughly, with many genes already identified. Key papers on seed coat color in P. vulgaris need to be covered in this review.

Response: Thank you very much. There is a table in our paper. This paper only discusses about the seed coat colour progress in adzuki bean. The genetic mechanism of common bean grain colour is significantly different from that of adzuki bean, especially in mottled colours. So, it is difficult to discuss with comparison.

Many more corrections, suggestions and questions are available in the paper file (see attached). I hope these might be helpful. Please note that my comments on language and style are not exhaustive, there should be a serious work on text revision.

Response: Thank you very much. We have improved our manuscript as your suggestion.
I recommend to reject this paper with an encouragement to resubmit it after very deep elaboration. I also wish authors good luck.

Reviewer 2 Report

The review article is interesting and shows that state of the art. It is useful for researchers and students related to the topic.

However, to improve the quality of the article, it is important to correct:

i) Tables and Figures ought to be clearly explained, giving enough information to the reader, who needs to understand them without reading the whole text or the section where the Figure/Table is citated; that is why Figure/Table has a Title and an explanatory legend, including the most important information than the authors consider to highlight.

ii) Small corrections throughthe text. 

Please, see file attached.

Author Response

The review article is interesting and shows that state of the art. It is useful for researchers and students related to the topic.

However, to improve the quality of the article, it is important to correct:

  1. i) Tables and Figures ought to be clearly explained, giving enough information to the reader, who needs to understand them without reading the whole text or the section where the Figure/Table is citated; that is why Figure/Table has a Title and an explanatory legend, including the most important information than the authors consider to highlight.

Response: Thank you very much. We have tried to explain the Tables and Figures more clearly.

  1. ii) small corrections through the text. 

Please, see file attached.

Response: Thank you for your patience. We have corrected issues in our article as your suggestion.

Reviewer 3 Report

Comments to the manuscript plants-2500551 "Progress in adzuki bean seed coat colour studies".

Authors propose a good review of the bibliography on the knowledge of the genetic determinism of the adzuki bean coat colour. The manuscript is well organized and written, with some clear and informative figures. In my opinion, it is suitable for publication after some minor changes.

1) Firstly authors are invted to use the citation system of the Journal according to the author's guide.

2) Lines 56 and 65: the article reported in the reference list is "Jin,1996" not "Jin et al., 1996";

3) Lines 75-76: Please check the text is interrupted;

4) Lines 123-124: Please check, in the reference list two articles are cited as Chu et al., 2021a and 2021b; what is the right citation?

5) Lines 394-395: the article of Rischkowsky and Pilling, 2007 was not cited in the text;

6) Lines 412-415: two articles are reported in the reference list as Wang et al. 2022 but only one was cited in the text.

The English language is fine. Only minor editing changes are required.

Author Response

Comments to the manuscript plants-2500551 "Progress in adzuki bean seed coat colour studies".

Authors propose a good review of the bibliography on the knowledge of the genetic determinism of the adzuki bean coat colour. The manuscript is well organized and written, with some clear and informative figures. In my opinion, it is suitable for publication after some minor changes.

Response: Thank you for your patience. We have corrected issues in our article.

1) Firstly, authors are invited to use the citation system of the Journal according to the author's guide.

Response: Thank you very much. We have checked the citations in our article.

2) Lines 56 and 65: the article reported in the reference list is "Jin,1996" not "Jin et al., 1996";

Response: Thank you very much. We have corrected this mistake.

3) Lines 75-76: Please check the text is interrupted;

Response: Thank you very much. We have deleted this interrupted.

4) Lines 123-124: Please check, in the reference list two articles are cited as Chu et al., 2021a and 2021b; what is the right citation?

Response: Thank you very much. The two references are both right. The 2021a and 2021b were used to distinguish these two references.

5) Lines 394-395: the article of Rischkowsky and Pilling, 2007 was not cited in the text;

Response: Thank you very much. We have deleted this reference.

6) Lines 412-415: two articles are reported in the reference list as Wang et al. 2022 but only one was cited in the text.

Response: Thank you very much. We have deleted the wrong one.

Round 2

Reviewer 1 Report

As I may see, the authors have conducted some revision on their manuscript. Unfortunately, many of the suggestions made during the first revision were not taked into account.

1. Both language and style still require improvement. The authors seem to confirm my corrections but these are not enough.
2. The authors state that 'Figure 2 clearly explained the genetic relationship of eight adzuki bean seed coat colour in our paper before'. Actually, it did not and remains intact. The sentence which was added to a figure caption does not help. It is unclear what locus corresponds to 'grey with black mottle' phenotype and why arrows from 'black' and 'grey with black mottle' converge together in 'light brown'. It is very difficult to understand why these arrows sometimes brach and sometimes unite with each other. The authors mention interactions like dominant or recessive epistasis in their text; however, statements like 'genes that regulate these epistasis situations' are not easy to understand, as genes regulate some processes, not modes of interaction.
I recommend to reformat the authors' Fig. 2 following the format of Fig. 1 from this paper:
https://www.science.org/doi/10.1126/science.aao3526
Each phenotype there is followed by its genotype. For example, there may be seed photographs from the original Fig. 2 with captions like ivory (bb rbrb tt yy brbr gg rr), light brown (bb T_ R_) etc. (I am not sure if I reconstructed the interactions between genes correctly). Moreover, in Fig. 1 in the recommended paper there is a scheme of interactions with arrows, enzymes and substances. As some of the genes/pigments in adzuki bean have not been identified yet, there may be gene names instead of enzymes and some abstract characteristics (like 'black pigment' or 'no pigments') instead of formulas. Without the significant improvement of genetic part this paper cannot be recommended for publication.
3. The title of Table 1 needs to be elaborated. It is not clear what authors mean by 'genetic efforts' and how these efforts could be represented in a table. The note under this table is excessive: as you have already referenced some sources, it is clear that this data is from previously published papers.
Please italicize name of loci. There is no need to start each name of a color with a capital letter. Please also reformat a column entitled 'Chromosome/Linkage group'. Words 'Linkage Groups' should be removed from each cell. LG numbers are usually given as Roman, not Arabic, figures (I, IV, VII etc.). I am sure there is a clear idea of correspondence between LGs and chromosomes, so I suggest that the authors should replace 'Chromosome 3' with the corresponding LG.
4. It is unclear why some parts of formulas in Fig. 3C-D are colored with red, blue or green (there is no explanation in this figure's caption) but there are many more minor concerns which are too early to discuss.

There should be at least one more round of revision after the language/style correction and deep elaboration of genetic part of the paper.  

There should be much more improvement of this paper's language and style.

Author Response

As I may see, the authors have conducted some revision on their manuscript. Unfortunately, many of the suggestions made during the first revision were not taked into account.

  1. Both language and style still require improvement. The authors seem to confirm my corrections but these are not enough.

Response: Thank you very much. We have improved the manuscript.

  1. The authors state that 'Figure 2 clearly explained the genetic relationship of eight adzuki bean seed coat colour in our paper before'. Actually, it did not and remains intact. The sentence which was added to a figure caption does not help. It is unclear what locus corresponds to 'grey with black mottle' phenotype and why arrows from 'black' and 'grey with black mottle' converge together in 'light brown'. It is very difficult to understand why these arrows sometimes brach and sometimes unite with each other. The authors mention interactions like dominant or recessive epistasis in their text; however, statements like 'genes that regulate these epistasis situations' are not easy to understand, as genes regulate some processes, not modes of interaction.
    I recommend to reformat the authors' Fig. 2 following the format of Fig. 1 from this paper:
    https://www.science.org/doi/10.1126/science.aao3526
    Each phenotype there is followed by its genotype. For example, there may be seed photographs from the original Fig. 2 with captions like ivory(bb rbrb tt yy brbr gg rr), light brown(bb T_ R_) etc. (I am not sure if I reconstructed the interactions between genes correctly). Moreover, in Fig. 1 in the recommended paper there is a scheme of interactions with arrows, enzymes and substances. As some of the genes/pigments in adzuki bean have not been identified yet, there may be gene names instead of enzymes and some abstract characteristics (like 'black pigment' or 'no pigments') instead of formulas. Without the significant improvement of genetic part this paper cannot be recommended for publication.

Response: Thank you very much. We have rebuilt the Figure 2. The Figure 2 was only used to explain the genetic relationship of different seed coat colour in adzuki bean without enzymes and substances. We hope you can find the new Figure 2 easier to understand.
3. The title of Table 1 needs to be elaborated. It is not clear what authors mean by 'genetic efforts' and how these efforts could be represented in a table. The note under this table is excessive: as you have already referenced some sources, it is clear that this data is from previously published papers.
Please italicize name of loci. There is no need to start each name of a color with a capital letter. Please also reformat a column entitled 'Chromosome/Linkage group'. Words 'Linkage Groups' should be removed from each cell. LG numbers are usually given as Roman, not Arabic, figures (I, IV, VII etc.). I am sure there is a clear idea of correspondence between LGs and chromosomes, so I suggest that the authors should replace 'Chromosome 3' with the corresponding LG.

Response: Thank you very much. We have improved the Table 1 as you wish. But the ‘Chromosome 3’ could not be replaced by LG. The LG may be replaced by the improved result ‘Chromosome’. But the LG and Chromosome result does not come from the same team. So, there is not a clear idea of correspondence between LGs and chromosomes.
4. It is unclear why some parts of formulas in Fig. 3C-D are colored with red, blue or green (there is no explanation in this figure's caption) but there are many more minor concerns which are too early to discuss.

Response: Thank you very much. Red, blue or green in Fig. 3C-D represent ring

system of the metabolite.

Round 3

Reviewer 1 Report

As I may see, the authors have revised their manuscript quite thoroughly. Its language and style have been improved although there are still some points to be corrected. Some minor language issues can be improved in the course of this paper's preparation for publishing. However, some of the unclear statements can be corrected only by the authors. Still, the genetic terminology requires improvement. There cannot be situations like 'light brown is dominant to ivory seed coat colour, which are controlled by two loci' as dominance is a type of interactions between alleles of one gene, not the case of several genes.
Please italicize genes' names and abbreviations including the updated Figure 2. In its current form this figure is indeed better for understanding. However, dominant alleles influence a phenotype even in a heterozygous state. It means that, say, the genotype of black-seeded cultivar should look like R_ rbrb brbr yy T_ gg B_. Of course, a cultivar is most likely homozygous but here you represent possible genotypes corresponding to certain phenotypes.
Again, in Table 1, please (1) italicize loci names in the first column, (2) replace capitalized letters from the second column, and (3) replace 'Linkage Groups' (why plural?) in the third column with 'LG'. Please also note that in the 'Reference' column you need to format references according to this journal's guide for authors.
Please also double-check all chemical nomenclature. For example, in line 746, 'catechinopyranopyranopyranopyranocyanidins' is clearly an error. I have also found when looking up in Google Scholar that a term 'pionidin' is almost never used. Instead, it is usually spelled as 'peonidin' or (more rarely) 'paeonidin', both referring to Paeonia, the plant this pigment was first extracted from.
The process of eradication of minor flaws from this manuscript might continue but let reads themselves assess the quality of this paper. After the suggestions made here are considered, this paper can be accepted for publication.

Some language check is needed, especially in what refers to terms.

Author Response

As I may see, the authors have revised their manuscript quite thoroughly. Its language and style have been improved although there are still some points to be corrected. Some minor language issues can be improved in the course of this paper's preparation for publishing. However, some of the unclear statements can be corrected only by the authors.

Response: Thank you very much. We have improved the manuscript.

Still, the genetic terminology requires improvement. There cannot be situations like 'light brown is dominant to ivory seed coat colour, which are controlled by two loci' as dominance is a type of interactions between alleles of one gene, not the case of several genes.

Response: Thank you very much. But as we know that ‘dominant’ can also be used to describe the relationship between two qualitative traits including seed coat colour. We have also improved the genetic terminology as your suggestion.

Please italicize genes' names and abbreviations including the updated Figure 2. In its current form this figure is indeed better for understanding. However, dominant alleles influence a phenotype even in a heterozygous state. It means that, say, the genotype of black-seeded cultivar should look like R_ rbrb brbr yy T_ gg B_. Of course, a cultivar is most likely homozygous but here you represent possible genotypes corresponding to certain phenotypes.

Response: Thank you very much. We have improved the Figure 2 as your suggestion

Again, in Table 1, please (1) italicize loci names in the first column, (2) replace capitalized letters from the second column, and (3) replace 'Linkage Groups' (why plural?) in the third column with 'LG'. Please also note that in the 'Reference' column you need to format references according to this journal's guide for authors.

Response: Thank you very much. We have improved the Table 1 as your suggestion and format the references according to this journal's guide for authors.

Please also double-check all chemical nomenclature. For example, in line 746, 'catechinopyranopyranopyranopyranocyanidins' is clearly an error. I have also found when looking up in Google Scholar that a term 'pionidin' is almost never used. Instead, it is usually spelled as 'peonidin' or (more rarely) 'paeonidin', both referring to Paeonia, the plant this pigment was first extracted from.

Response: Thank you very much. We are sorry for the misspelling of chemical nomenclature. We have replaced them by the right spelling and rechecked their name.

The process of eradication of minor flaws from this manuscript might continue but let reads themselves assess the quality of this paper. After the suggestions made here are considered, this paper can be accepted for publication.